# Newly Reported Studies on the Increase in Gastrointestinal Symptom Prevalence withCOVID-19 Infection: A Comprehensive Systematic Review and Meta-Analysis

**DOI:** 10.3390/diseases8040041

**Published:** 2020-11-10

**Authors:** Hakan Akin, Ramazan Kurt, Fatih Tufan, Ahmed Swi, Resat Ozaras, Veysel Tahan, Ghassan Hammoud

**Affiliations:** 1Birinci International Hospital, Istanbul 34525, Turkey; drhakanakin@birincisaglikgrubu.com; 2Sondurak Medical Center, Istanbul 34764, Turkey; rmznkurt@yahoo.com; 3Independent Investigator, Istanbul 34107, Turkey; fatihtufan@gmail.com; 4Division of Gastroenterology & Hepatology, Department of Internal Medicine, University of Missouri, Columbia, MO 65212, USA; swia@health.missouri.edu (A.S.); hammoudg@health.missouri.edu (G.H.); 5Medilife International Hospital, Istanbul 34523, Turkey; rozaras@yahoo.com

**Keywords:** COVID-19, gastrointestinal symptom, systematic review, meta-analysis, diarrhea, nausea, vomit, abdominal pain

## Abstract

Background and Aim: Although constitutional and respiratory symptoms such as cough and fever are the most common symptoms in patients infected with COVID-19, gastrointestinal (GI) tract involvement has been observed by endoscopic biopsies. Multiple GI symptoms, including diarrhea, nausea or vomiting and abdominal pain, have also been reported. This review aims to present the currently available data regarding the GI symptoms of COVID-19 patients, and to compare the frequency of GI symptoms in early stage (Eastern) mostly Chinese data to the current stage (Western) non-Chinese data. Methods: We performed a systematic literature search to identify both published studies by using PubMed, Google Scholar, and CNKI (Chinese medical search engine), and yet unpublished studies through medRxiv and bioRxiv. We also reviewed the cross references of the detected articles. We conducted a Medical Subject Headings (MeSH) search up until 20 September 2020. We pooled the prevalence of symptoms of diarrhea, anorexia, nausea, vomiting, and abdominal pain by using the Freeman–Tukey’s transforming random effect model. Results: A total of 118 studies were included in the systematic review and 44 of them were included in the meta-analysis. There was a significant heterogeneity between the studies; therefore, the random effects model was used. The pooled prevalence estimate of any GI symptoms reported was found to be 0.21 (95%CI, 0.16–0.27). Anorexia was the most commonly reported GI symptom at 18% (95%CI, 0.10–0.27) followed by diarrhea at 15% (95%CI, 0.12–0.19). Diarrhea, abdominal pain, nausea/vomiting, and respiratory symptoms were more common in non-Chinese studies. The prevalence of abdominal pain was lower in the “inpatient-only” studies when compared with studies that included outpatients only and those including both inpatients and outpatients. Conclusions: In this comprehensive systematic review and meta-analysis study, we observed higher rates of diarrhea, nausea/vomiting, and abdominal pain in COVID-19 infected patients among non-Chinese studies compared to Chinese studies. We also observed a higher prevalence of GI symptoms in Chinese studies than was reported previously. Non-respiratory symptoms, including GI tract symptoms, should be more thoroughly and carefully evaluated and reported in future studies.

## 1. Introduction

Coronavirus disease 2019 (COVID-19) emerged in December 2019 in the Wuhan region of China and spread across the world [1]. The causative virus was subsequently named severe acute respiratory syndrome coronavirus 2 (SARS-CoV-2) [2]. The infection is spread by droplets, aerosols, and direct contact with contaminated surfaces. As of 2 October 2020, there have been over 33,842,281 confirmed cases of COVID-19, including 1,010,634 deaths, reported to WHO [3]. Although respiratory symptoms such as fever and cough are the most common findings in patients with COVID-19, GI tract involvement has been identified by endoscopic biopsies. In addition, the presence of several GI symptoms, including diarrhea, nausea/vomiting, and abdominal pain, has been reported [4]. COVID-19 uses an ACE2 protein to gain entry into cells [5]. ACE2 receptors are widely distributed in the human body, including in the lung, liver, stomach, ileum, colon, and kidney, allowing COVID-19 to infect many different organs.

Human coronaviruses are known to cause respiratory and enteric symptoms. During the SARS outbreak in 2002–2003, 16–73% of patients with SARS had diarrhea [6]. Early reports of COVID-19 may not represent the actual rate of GI symptoms, because in the early times of the outbreak the most notable symptoms were severe respiratory symptoms. Viral RNA has been detected in GI epithelial cells and stool of COVID-19 patients, pointing to a possible fecal–oral transmission of the SARS-CoV-2 infection [7]. This review aims to present the currently available data regarding GI symptoms of COVID-19 patients and to compare the frequency of GI symptoms in early stage (Eastern) to current stage (Western) data.

## 2. Methods

Literature search strategies: This study was designed to be a systematic review and meta-analysis. Literature for this study was identified by searching the following online databases in the English and Chinese languages: PubMed, Google Scholar, and CNKI (Chinese medical search engine); and MedRvix and BioRxiv for the yet unpublished manuscripts. Additionally, the cross references of the detected articles were screened. We did a Medical Subject Headings (MeSH) search for two terms:(1)Term 1: COVID-19(supplementary concept).(2)Term 2: Severe acute respiratory syndrome Coronavirus 2 (supplementary concept).

Besides those two MeSH terms, we added the terms gastrointestinal, digestive, diarrhea, nausea, vomiting, anorexia, abdominal pain, and abdominal discomfort symptoms to our search.

The inclusion criteria were:(1)Any study including data of “COVID-19 documented patients” which was defined as patients with relevant clinical manifestations and radiologic findings of COVID-19 infection, plus PCR and/or antigen test for COVID-19 infection positivity.(2)Any study reporting respiratory and GI symptoms of “documented COVID-19 infection” (as defined above) was included.(3)Authors that clearly reported the absence of GI symptoms were also included in this study.

The exclusion criteria were:(1)If the index study included non-documented or suspected COVID-19 infection data.(2)If the presence or absence of GI symptoms was not reported.(3)Non-scientific commentaries and reports, reviews, meta-analysis studies, and scientific news.

All studies between 01 December 2019 and 19 September 2020 were screened through the relevant literature and then detection of possibly related studies through a search of manuscript titles and abstracts was done. After selection of possibly concordant studies, a full text article evaluation was conducted. During this full text evaluation phase, we assessed whether the manuscript was fulfilling the inclusion and exclusion criteria. That phase was done by each of the three researchers (H.A., A.S. and R.K.) independently. The three sets of study selections were then compared. The disagreements between selected reports were resolved through discussion with the three other authors (F.T., R.O. and V.T.). After the articles were selected for systematic review, data were extracted by (H.A., A.S., R.K., F.T. and R.O.) to an Excel data sheet. During data extraction, for the parameters that included more than one subgroup, we counted patients who had more than one of the subgroups only once in that parameter. For instance, if the parameter was a symptom—which could have included multiple subgroups, such as respiratory symptoms, any GI symptom, nausea or vomiting, and abdominal pain or discomfort, we counted a patient who had nausea and abdominal pain only once in the symptoms parameter. In studies where the number of patients having symptoms was given for each symptom, but did not indicate how many patients had multiple symptoms, we chose the highest numbered symptom as the minimum total number of that parameter.

The type and definitions of data extracted to the Excel spread sheet are given below:Authors’ names;Title of the manuscript;Study language: no language restriction was done;Place of study: where the study population’s data were collected and not where they were published;Study date: beginning and end period of study period;Number of patients included: study population;Age (mean or median (± 1 SD or IQR); range);Age related type of the study: we separated the studies according to age groups since age data were significantly heterogeneous;Pediatric studies: 0–14 years;Mixed studies: studies including both children and adults, adult studies of 15 years or higher;Male: number and percentage of male subjects included in the study

Severe disease* (*n*, %): We modified and used the Chinese definition of COVID-19 severity from “Diagnosis and Treatment Plan for Pneumonia Infected by New Coronavirus (Trial Sixth Edition)” National Health Commission of the People’s Republic of China [8]. We combined the severe plus critically ill group plus deaths then recoded as “severe COVID-19 disease.” The other severity definition was a modified definition (mild and moderate cases) recoded as “non-severe COVID-19 disease.” The Chinese definition of severe disease and critical disease is as follows [8];

Severe cases: Adult cases meeting any of the following criteria: (1) Respiratory distress (≥ 30 breaths/min). (2) Oxygen saturation ≤93% at rest. (3) Arterial partial pressure of oxygen (PaO_2_)/fraction of inspired oxygen (FiO2) ≤300 mmHg (l mmHg = 0.133 kPa). In high-altitude areas (at an altitude of over 1000 m above the sea level), PaO_2_/FiO_2_ shall be corrected by the following formula: PaO2/FiO_2_x (Atmospheric pressure (mmHg)/760). Cases with chest imaging that show obvious lesion progression within 24–48 h >50% shall be managed as severe cases. Child cases meeting any of the following criteria: (1) Tachypnea (RR ≥60 breaths/min for infants aged below 2 months; RR ≥ 50 BPM for infants aged 2–12 months; RR ≥40 BPM for children aged 1–5 years; and RR ≥ 30 BPM for children above 5 years old) independent of fever and crying. (2) Oxygen saturation ≤92% on finger pulse oximeter taken at rest. (3) Labored breathing (moaning, nasal fluttering, and infrasternal, supraclavicular, and intercostal retraction), cyanosis, and intermittent apnea. (4) Lethargy and convulsion. (5) Difficulty feeding and signs of dehydration.

Critical cases: Cases meeting any of the following criteria: 4.1 Respiratory failure and requiring mechanical ventilation; 4.2 Shock; 4.3 with other organ failure that requires ICU care.

Fever (*n*, %): number and percentage of patients having clinical fever.

Respiratory symptoms (*n*, %): Number and percentage of patients having any type of respiratory symptoms, including cough, dyspnea, expectoration, and any other reported respiratory symptoms. Here we combined the respiratory symptoms and got one highest patient numbered symptom item as minimum number of “respiratory symptom” parameter.

Presence of any GI symptoms (*n*, %): number and percentage of patients having any type of GI symptoms, including anorexia, nausea/vomiting, diarrhea, and abdominal pain/discomfort. As some patients may have more than one symptom at the same time, we counted each as one even if the patient had multiple symptoms. On the other hand, we combined the symptoms and made the highest number of patient symptoms as the minimum number of “any GI symptom” parameter.

Anorexia (*n*, %): number and percentage of patients having anorexia symptom.

Nausea/vomiting (*n*, %): Number and percentage of patients having nausea and/or vomiting symptom. Here we combined the numbers and got the highest number of patient symptoms as the minimum number of “nausea/vomiting” parameter.

Diarrhea (*n*, %): number and percentage of patients having any type of diarrhea symptom.

Abdominal pain/discomfort (*n*, %): Number and percentage of patients having abdominal pain and/or abdominal discomfort symptoms. Here we combined the numbers and made the highest number of patient symptoms as the minimum number of the “abdominal pain/discomfort” parameter.

After the Excel data sheet was fulfilled, data were then transformed to the statistical program by the researchers H.A., A.S., R.K., F.T. and R.O.

## 3. Statistical Analysis

The prevalence and 95% confidence interval (CI)values of diarrhea, nausea/vomiting, abdominal pain, and anorexia symptoms were estimated using the meta-analysis package of the R program (4.14-0 Meta package, http://cran.r-project.org/web/packages/meta/meta.pdf). The Freeman–Tukey’s transforming random effect model was used for the analyses. I^2^ values were used to measure heterogeneity between the studies. Chinese studies were compared with non-Chinese studies, and “inpatient-only” studies were compared with studies including any outpatients, i.e., “mixed studies” reporting the data of inpatients together with outpatients as clinical setting plus outpatient-only. Using sub-group analysis and random effects, the differences between the groups were assessed.

## 4. Results

A total of 1050 full text studies were screened and 205 potential studies were evaluated in detail. Among these, 118 studies were included in the systematic review and 44 were included in the meta-analysis. The flow chart of the study is shown in Figure 1.

Table 1 summarizes the characteristics of the studies included in the systematic review. Most of the studies reported the rate of diarrhea (105/118), while they less commonly reported the other GI symptoms (nausea 74/118, abdominal pain 39/118, and anorexia 38/118).

Among the studies, 92 took place in China and the remaining 26 took place in other countries. The pooled prevalence of diarrhea, abdominal pain, nausea/vomiting, and respiratory symptoms was significantly more common in non-Chinese studies when compared with Chinese studies (Table 2). The occurrences of other study variables in Chinese and non-Chinese studies were similar.

While most of the studies included “inpatient-only” (*n* = 81), 34 of them included both inpatients and outpatients, and two comprised “outpatient-only.” One study did not report inpatient or outpatient status of the included patients. The pooled prevalence of abdominal pain was significantly lower in “inpatient-only” studies. We could only compare these “inpatient-only” studies with those called mixed studies, i.e., inpatients together with outpatients (Table 3).

In this study’s evaluation phase, we detected four “outpatient-only” studies out of 205 eligible studies. Since there were only 951 patients from the “outpatient-only” studies, it was not suitable to compare those with “inpatient-only” studies. Thus, we performed a separate meta-analysis for the four “outpatient-only” studies. The prevalence of GI symptoms was as follows in the “outpatient-only” studies: diarrhea 0.10 (95% CI, 0.01, 0.27), any GI symptoms 0.13 (95% CI, 0.01, 0.41), respiratory symptoms 0.19 (95% CI, 0.07, 0.36), fever 0.63 (95% CI, 0.27, 0.93), and severe COVID-19 disease 0.08 (95% 0.01, 0.18).

The studies included in the meta-analysis had significant heterogeneity (I^2^ = 99%, *p*< 0.001); therefore, the random effects model was taken into consideration. The meta-analysis revealed that the most commonly reported GI symptoms were anorexia followed by diarrhea, whereas the least common symptom was abdominal pain. Forest plot graphics for diarrhea, nausea/vomiting, abdominal pain, and anorexia are shown in Figure 2, Figure 3, Figure 4 and Figure 5, respectively.

## 5. Discussion

The novel SARS-CoV-2 is currently causing a major pandemic that constitutes a world health crisis. COVID-19 patients commonly have fever and respiratory illness. However, some patients also complain of GI symptoms such as diarrhea, nausea/vomiting, and abdominal pain. Fecal–oral transmission of COVID-19 infection has been confirmed by the fact that the virus can replicate in both the respiratory and digestive tracts [123]. The first step of viral entry into enterocytes occurs via angiotensin-converting enzyme 2 (ACE2) binding to ACE2 receptors on the surfaces of enterocytes, similarly to SARS-CoV [4]. After entry into the host cell, viral RNA and proteins are produced by ribosomes. Viral capsids, RNA, and proteins combine to form multiple new copies of COVID-19. These viral particles lead to cytokine release (interleukin (IL)-2, IL-7, tumor necrosis factor (ΤΝF)-α, and macrophage and monocyte products), which mediate various effects on organs. The virus can then spread to other digestive organs, such as the liver using the same ACE2 enzyme [124].

Patients with metabolic conditions such as obesity, diabetes, cardio-metabolic problems, and liver diseases were repeatedly reported to have higher rates of COVID-19 related morbidity in various studies [17,19,125].Gut microbiota can influence the immune response via affecting disease progression. Not only over-active, but also a hypo-active immune response possibly mediated by gut microbiota may lead to severe clinically adverse events. The colon includes a large density of bacteria in the families of Bacteroidaceae, Prevotellaceae, Rikenellaceae, Lachnospiraceae, and Ruminococcaceae [126], while Bacteroidetes, Firmicutes, and Proteobacteria are more preponderate in the lung [127]. The gut microbiota may affect pulmonary health through interactions between the gut microbiota and the lungs, named the “gut–lung axis” [128]. The gut–lung axis is reciprocal, so endotoxins, microbial metabolites, can affect the lung through the blood and inflammation of the lungs can affect the gut microbiota as well [129]. Several studies have shown that respiratory infections are associated with a change in the composition of the gut microbiota [130]. Multiple data suggest that the gut microbiota play a key role in the pathogenesis of sepsis and Adult Respiratory Distress Syndrome (ARDS). Loss of gut bacterial diversity may lead to dysbiosis that can be associated with many diseases [131]. As many elderly and immune-compromised patients progress to serious adverse clinical consequences in Covid-19, possible cross-talk may be occurring between the lung and intestinal microbiota, which may affect the outcome of the disease’s course.

To the best of our knowledge, this systematic review and meta-analysis presents the largest patient population involving COVID–19 infection and GI symptoms. We used prepublication repositories medRxiv and bioRxiv that enabled us to search and include unpublished manuscripts from gray literature and enlarge our study population drastically. Most of the studies outside of China began reporting in May 2020, especially those from Western countries. We included 31 Chinese studies with a total of 12,798 patients and 13 non-Chinese studies with a total of 50,094 patients. Therefore, we were able to effectively compare Chinese and non-Chinese COVID-19 infection studies according to their GI manifestations. Currently, the majority of original studies, systemic reviews, and meta-analysis studies are from China with a few exceptions and are mostly limited to the “inpatient-only” clinical setting [132]. We included clinical settings of both mixed studies (“inpatient together with outpatient” plus “outpatient-only”) and “inpatient-only” studies.

### 5.1. Diarrhea

In our meta-analysis the overall prevalence of diarrhea was 15%. The rate of diarrhea was significantly higher in non-Chinese studies (24%) compared to Chinese studies (12% and *p* < 0.001). Similar but lower prevalence rates have been reported in a recent meta-analysis [132]. The pooled prevalence estimate (PPE) of diarrhea was 7.7% in the overall population, 18.3% in non-Chinese studies, and 5.8% in Chinese studies.

In this study, the occurrence of diarrhea was similar for the “inpatient-only” study group (14%) and mixed patient study group (16% and *p* = 0.795). The meta-analysis by Sultan et al. [132] included 39 “inpatient-only” studies with a total of 8521 patients, and the PPE for diarrhea was 10.4%, which was lower than the corresponding PPE of 14.4% in this study.

We detected a total of four studies (three Chinese and one USA study), which tailored the clinical setting as “outpatient-only.” Since there was a low number of patients (*n* = 951) we could not compare the “outpatient-only” study group with the “inpatient-only” study group. However, a meta-analysis for these four “outpatient-only” studies was conducted. The PPE of diarrhea was 10%. On the other hand, the meta-analysis by Sultan et al. [132] analyzed three “outpatient-only” studies, including 1701 patients. In that meta-analysis, the PPE value for diarrhea in “outpatient-only” studies was 4%.

In a recent and large comprehensive meta-analysis about general evaluations of patients with COVID-19 infection included 59,254 patients mostly from China but also including 10 other countries, reported a PPE of 9% for any type of GI symptoms [125]. However, subgroups of GI symptoms were not given in that meta-analysis.

The inception studies from China mostly reported low incidence of diarrhea and other GI symptoms [120]. In an earlier meta-analysis including 6686 patients from 35 Chinese studies, which compromised “inpatient-only” studies, a PPE value of 9% for diarrhea was reported [133]. In contrast with low prevalence reports, in a study from Taizhou, China and another one from Shanghai, China in which 212 mild COVID-19 patients were included, the rates of any GI symptoms were reported as high as 42.8% and 43.8%, respectively [40,63]. More recent reports from China claimed even higher rates of diarrhea (49.5%) and any GI symptoms (79%) [35]. Interestingly, a Chinese study including 232 hospitalized patients from Wuhan compared the rate of diarrhea between a group of patients who were admitted from January 19 to February 11, 2020 with that of another non-overlapping group of patients who were admitted from February 12 to March 2020 at the same hospital setting [43]. They concluded that as the COVID-19 infection outbreak progressed, the rate of diarrhea increased from 19% up to 43% (*p* = 0.022) in these two distinct groups of hospitalized patients.

The increase in the reported rates of GI symptoms from Chinese studies needs further explanation. Possible explanations may include increased awareness of non-respiratory symptoms, increased documentation, and re-infection as the outbreak progressed. Re-infection is common for “seasonal” coronaviruses 229E, OC43, NL63, and HKU1 [134]. COVID-19 can also reoccur after the first infection. It was confirmatively reported that one patient had a re-infection instead of persistent viral shedding from the first infection, by the epidemiological, clinical, serological, and genomic analyses [135]. These results showed that SARS-CoV-2 may continue to circulate among human populations.

### 5.2. Nausea/Vomiting

The PPE of nausea/vomiting for the whole study population was 10%, which included 44 studies with a total of 46,390 patients. The PPE of nausea/vomiting was significantly higher in non-Chinese studies (17%) than in Chinese studies (7% and *p* < 0.001).

In a recent meta-analysis by Sultan et al. [132], the overall PPE of nausea/vomiting was reported to be 7.8%. It was also noted that nausea/vomiting PPE in non-Chinese studies had a higher value (14.9%) than Chinese studies (5.2%). These results are in line with the current study results but have lower rates.

In this study, we reported the PPE for of nausea/vomiting for 22 “inpatient-only” studies was (12%). The PPE was (6%) for nausea/vomiting in an earlier Chinese meta-analysis that included 6686 hospitalized patients from 35 studies [133]. In this study, PPE of nausea/vomiting in the 22 mixed studies was (8%). Still, the difference was not statistically significant between “inpatient-only” and mixed-study groups.

### 5.3. Abdominal Pain/Discomfort

In the whole study population, the overall PPE of the abdominal pain/discomfort symptom was 6%. The abdominal pain/discomfort symptom was significantly higher in non-Chinese studies (9%), compared to Chinese studies (4% and *p* < 0.001). In a recent meta-analysis by Sultan et al. (133), the overall PPE of the abdominal pain/discomfort symptom was reported as 3.6%. Sultan et al. reported that non-Chinese studies had a PPE value of 5.3%, which is higher than that of Chinese studies (2.7%) and is in line with our results, but still has lower rates than what we have seen. The occurrence of the abdominal pain or discomfort symptom was significantly higher in the mixed-patient study group (7%) when compared to the “inpatient-only” study group (4% and *p*= 0.032).

### 5.4. Anorexia

The overall PPE of anorexia was 18% in the whole study population and was the most prevalent GI symptom. The anorexia symptom was higher in Chinese studies (21%) when compared to non-Chinese studies (17%), but the difference was not statistically significant (*p*: 0.783). In a Chinese meta-analysis including 6686 patients, the reported PPE of the anorexia symptom was the same as that in this study, 21% [133]. Usage of experimental drugs and herbal medicine against COVID–19 infections might be a possible explanation for the higher rate of anorexia symptom PPE values [136].

### 5.5. Limitations

In the aftermath of this pandemic, there was a skyrocketing of COVID-19 infection-related publications, which certainly led to a compromised adherence to some scientific publication criteria. Though at a time of disaster, these issues were criticized, and concerns about the quality of publications were raised [137]. The nature of such a big disaster would probably be associated with potential bias. In the related literature, we observed that the patient cohorts included very heterogeneous groups. We tried our best to prevent doubling of patients and we excluded some manuscripts to prevent overlapping. In most of the studies, the authors did not clearly state whether they did a full GI system evaluation upon admission. For most of the studies, the time periods of the GI symptoms and latency of respiratory symptoms and fever was not stated. Most of the studies did not state the underlying systemic diseases, including GI diseases that may have affected the assessment of GI symptoms. Additionally, for most of the studies, treatment strategies, including drug and herbal medicines in Chinese studies, were not well documented. The severity of COVID-19 disease was not well defined with different studies using different definitions. Still, the presence of GI symptoms and the degree of severity of COVID-19 infection were not well documented in most of the studies. Since liver function tests were not mentioned in most of the studies, we did not scan laboratory liver abnormalities. Another limitation is the fact that most of the included studies did not report the rate of GI symptoms in subgroups of severe or non-severe patients or between other subgroups. Therefore, we could not perform a head-to-head comparison in this regard. Lastly, there were very few “outpatient-only” studies and we could not compare them specifically with “inpatient-only” studies.

## 6. Conclusions

In this highly populated and comprehensive systematic review and meta-analysis study, we reported high PPE rates of anorexia, diarrhea, nausea/vomiting, and abdominal pain. Although healthcare providers and patients are well aware of the common symptoms of COVID-19 such as fever, cough, and shortness of breath, there is need to raise the awareness about the fact that not all individuals present with these symptoms, as gastrointestinal symptoms are relatively common with this disease. The PPE rates of diarrhea, nausea/vomiting, and abdominal pain were significantly higher in non-Chinese studies compared to Chinese studies. We also observed a higher prevalence for GI symptoms in the Chinese studies than what was reported previously. Non-respiratory symptoms, including those related to the GI tract, should be more carefully evaluated and reported in future studies.

## Figures and Tables

**Figure 1 diseases-08-00041-f001:**
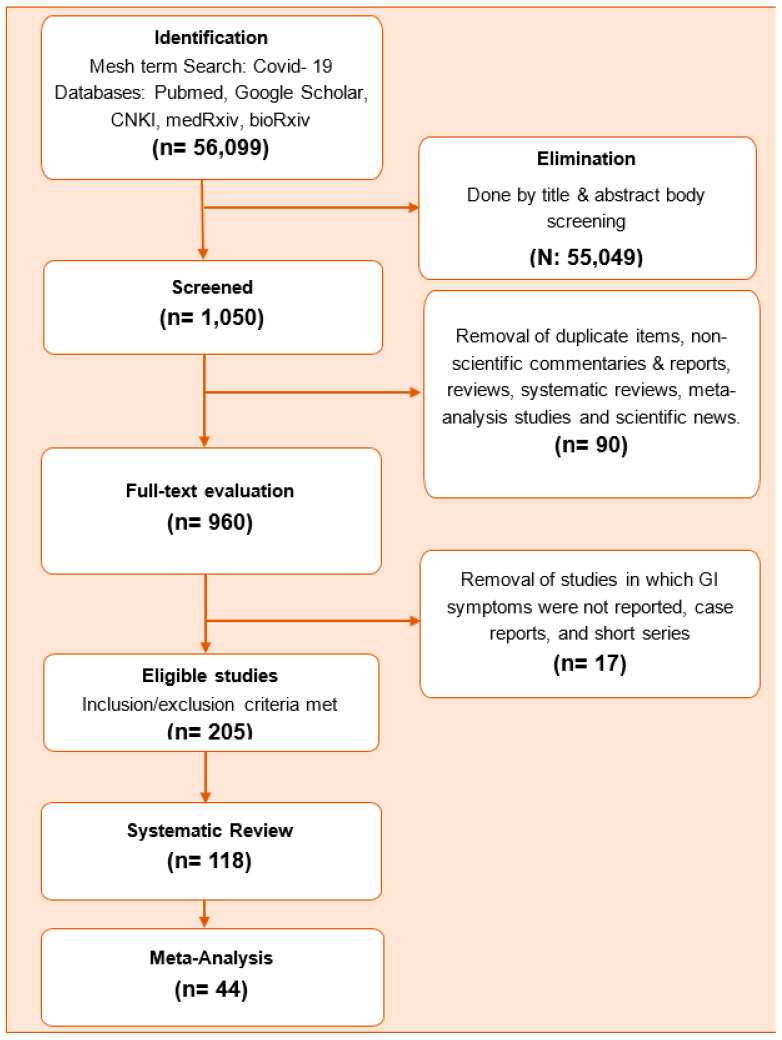
Study flow chart.

**Figure 2 diseases-08-00041-f002:**
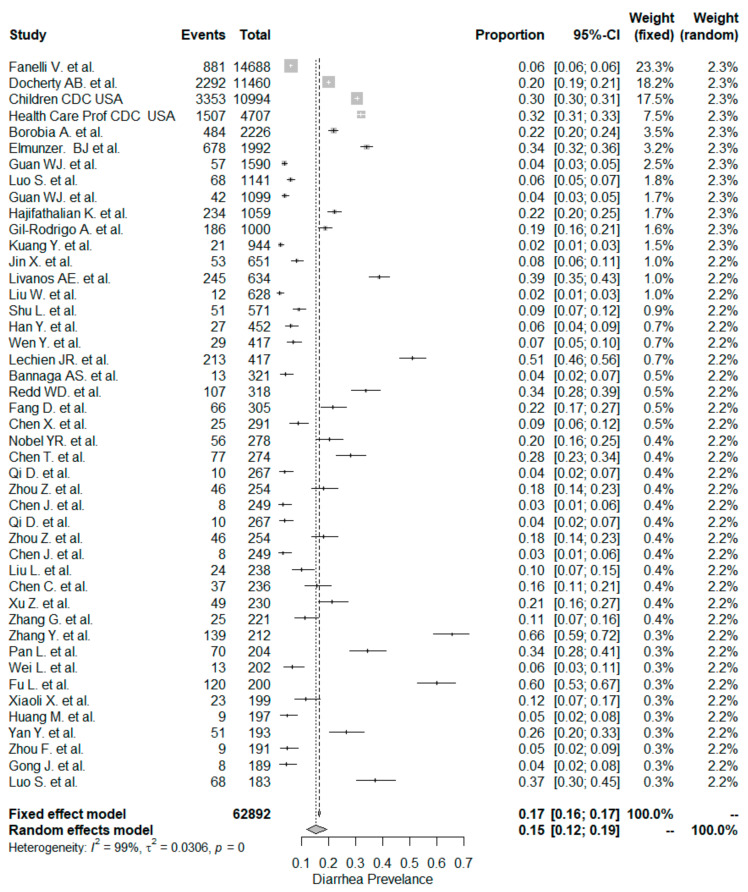
All studies, diarrhea forest plot and meta-analysis results.

**Figure 3 diseases-08-00041-f003:**
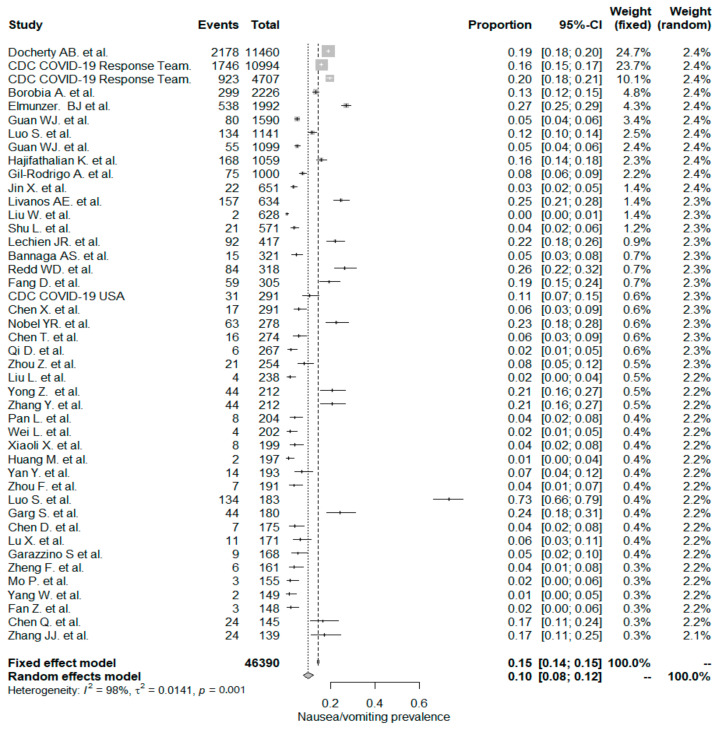
All studies, nausea/vomiting forest plot and meta-analysis results.

**Figure 4 diseases-08-00041-f004:**
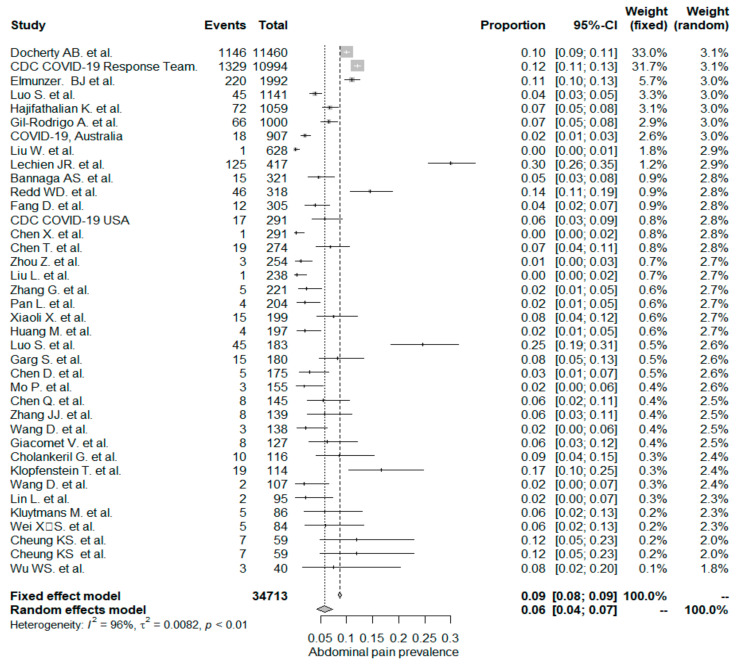
All studies, abdominal pain/discomfort forest plot and meta-analysis results.

**Figure 5 diseases-08-00041-f005:**
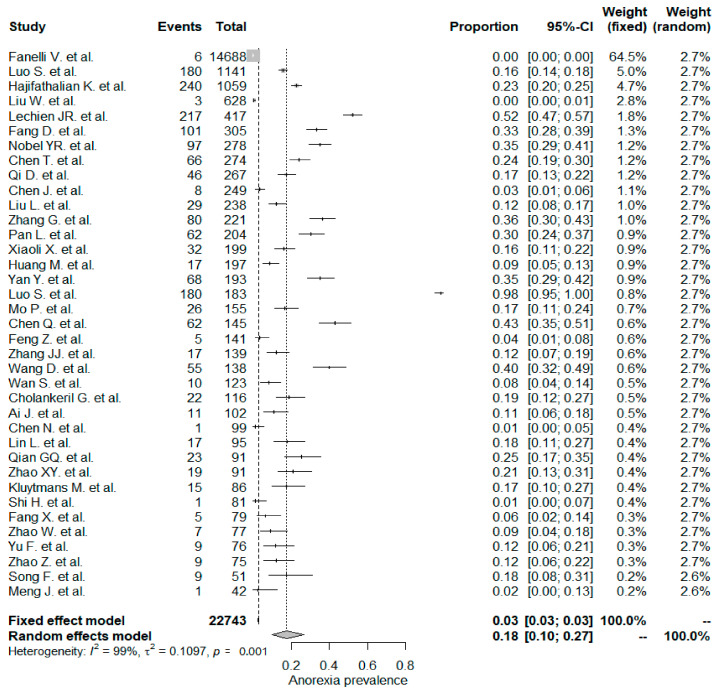
All studies, anorexia forest plot and meta-analysis results.

**Table 1 diseases-08-00041-t001:** Characteristics of the studies included in the systematic review and meta-analysis.

Authors	Study Type	Setting	Age Group	Place of Study	Number of Patients Included	Male Patients, *n* (%)	Severe Disease, *n* (%)	Diarrhea, *n* (%)	Nausea, *n* (%)	Abdominal Pain, *n* (%)	Anorexia, *n* (%)
Docherty AB. et al. [9]	RCS	Mixed	Mixed	UK	16,749	7715 (46, 06)	5528 (33)	2292 (13, 68)	2178 (13)	1146 (6, 84)	n.a.
Fanelli V. et al. [10]	RCS	IP	Mixed	Italy	14,688	9253 (63)	14,688 (100)	881 (6)	n.a.	n.a.	6 (0, 04)
CDC USA [11]	RCS	Mixed	adult	USA	10,994	5827 (53)	495 (4, 5)	3353 (30, 5)	1746 (15, 88)	1329 (12, 09)	n.a.
CDC USA [12]	RCS	Mixed	adult	USA	6760	2464 (36, 45)	184 (2, 72)	1507 (22, 29)	923 (13, 65)	n.a.	n.a.
Rojo JMC. et al. [13]	RCS	IP	adult	Spain	6424	3651 (56, 83)	2025 (31, 52)	n.a.	n.a.	n.a.	n.a.
Borobia A. et al. [14]	RCS	IP	adult	Spain	2226	1074 (48, 25)	460 (20, 66)	484 (21, 74)	299 (13, 43)	n.a.	n.a.
Elmunzer. BJ et al. [15]	RCS	IP	adult	USA and Canada	1992	1128 (56, 63)	646 (32, 43)	678 (34, 04)	538 (27, 01)	220 (11, 04)	n.a.
Guan WJ. et al. [16]	RCS	Mixed	adult	China	1590	904 (56, 86)	99 (6, 23)	57 (3, 58)	80 (5, 03)	n.a.	n.a.
Luo S. et al. [17]	CS	IP	adult	China	1141	n.a.	n.a.	68 (5, 96)	134 (11, 74)	45 (3, 94)	180 (15, 78)
Guan WJ. et al. [1]	RCS	Mixed	adult	China	1099	637 (57, 96)	173 (15, 74)	42 (3, 82)	55 (5)	n.a.	n.a.
Cheng JL. et al. [18]	CSS	IP	Mixed	China	1079	573 (53, 1)	72 (6, 67)	n.a.	n.a.	n.a.	n.a.
Hajifathalian K. et al. [19]	RCS	Mixed	adult	USA	1059	614 (57, 98)	n.a.	234 (22, 1)	168 (15, 86)	72 (6, 8)	240 (22, 66)
Gil-Rodrigo A. et al. [20]	RCS	Mixed	adult	Spain	1000	562 (56, 2)	119 (11, 9)	186 (18, 6)	75 (7, 5)	66 (6, 6)	n.a.
Kuang Y. et al. [21]	CSS	n.a.	adult	China	944	476 (50, 42)	n.a.	21 (2, 22)	n.a.	n.a.	n.a.
COVID-19, Australia [22]	ER	Mixed	Mixed	Australia	907	454 (50, 06)	17 (1, 87)	n.a.	n.a.	18 (1, 98)	n.a.
Jin X. et al. [23]	CSS	IP	adult	China	788	407 (51, 65)	78 (9, 9)	n.a.	n.a.	n.a.	n.a.
Jin X. et al. [24]	CCS	IP	adult	China	651	331 (50, 84)	205 (31, 49)	53 (8, 14)	22 (3, 38)	n.a.	n.a.
Livanos AE. et al. [25]	RCS	Mixed	adult	USA	634	369 (58, 2)	158 (24, 92)	245 (38, 64)	157 (24, 76)	n.a.	n.a.
Liu W. et al. [26]	CS	OP	adult	China	628	296 (47, 13)	75 (11, 94)	12 (1, 91)	2 (0, 32)	0 (0)	3 (0, 48)
Shu L. et al. [27]	RCS	IP	adult	China	571	278 (48, 69)	38 (6, 65)	51 (8, 93)	21 (3, 68)	n.a.	n.a.
Han Y. et al. [28]	RCS	Mixed	adult	China	452	254 (56, 19)	n.a.	27 (5, 97)	n.a.	n.a.	n.a.
Wen Y. et al. [29]	CSS	Mixed	adult	China	417	197 (47, 24)	36 (8, 63)	29 (6, 95)	n.a.	n.a.	n.a.
Lechien JR. et al. [30]	RCS	Mixed	adult	Europe	417	154 (36, 93)	0 (0)	213 (51, 08)	92 (22, 06)	125 (29, 98)	217 (52, 04)
Bannaga AS. et al. [31]	RCS	IP	adult	UK	321	189 (58, 88)	44 (13, 71)	13 (4, 05)	15 (4, 67)	15 (4, 67)	n.a.
Redd WD. et al. [32]	RCS	IP	adult	USA	318	174 (54, 72)	n.a.	107 (33, 65)	84 (26, 42)	46 (14, 47)	110 (34.8)
Fang D. et al. [33]	RCS	IP	adult	China	305	146 (47, 87)	46 (15, 08)	66 (21, 64)	59 (19, 34)	12 (3, 93)	101 (33, 11)
Chen X. et al. [34]	RCS	Mixed	Mixed	China	291	145 (49, 83)	50 (17, 18)	25 (8, 59)	17 (5, 84)	1 (0, 34)	n.a.
CDC USA [35]	RCS	Mixed	pediatric	USA	291	166 (57, 04)	4 (1, 37)	37 (12, 71)	31 (10, 65)	17 (5, 84)	n.a.
Nobel YR. et al. [36]	RCS	OP	adult	USA	278	145 (52, 16)	n.a.	56 (20, 14)	63 (22, 66)	n.a.	97 (34, 89)
Chen T. et al. [37]	RCS	IP	adult	China	274	171 (62, 41)	113 (41, 24)	77 (28, 1)	16 (5, 84)	19 (6, 93)	66 (24, 09)
Qi D. et al. [38]	RCS	IP	adult	China	267	149 (55, 81)	n.a.	10 (3, 75)	6 (2, 25)	n.a.	46 (17, 23)
Zhou Z. et al. [39]	RCS	IP	adult	China	254	115 (45, 28)	16 (6, 3)	46 (18, 11)	21 (8, 27)	3 (1, 18)	n.a.
Chen J. et al. [40]	RCS	IP	adult	China	249	126 (50, 6)	22 (8, 84)	8 (3, 21)	n.a.	n.a.	8 (3, 21)
Liu L. et al. [41]	RCS	Mixed	adult	China	238	138 (57, 98)	n.a.	24 (10, 08)	4 (1, 68)	1 (0, 42)	29 (12, 18)
Chen C. et al. [42]	RCS	IP	adult	China	236	110 (46, 61)	27 (11, 44)	37 (15, 68)	n.a.	n.a.	n.a.
Wan Y. et al. [43]	RCS	IP	Mixed	China	230	129 (56, 09)	61 (26, 52)	49 (21, 3)	n.a.	n.a.	n.a.
Yong Z.et al. [44]	RCS	Mixed	adult	China	212	85 (40, 09)	n.a.	93 (43, 87)	44 (20, 75)	n.a.	n.a.
Zhang Y. et al. [45]	CSS	IP	adult	China	212	85 (40, 09)	n.a.	139 (65, 57)	44 (20, 75)	n.a.	n.a.
Pan L. et al. [46]	CSS	IP	adult	China	204	107 (52, 45)	37 (18, 14)	70 (34, 31)	8 (3, 92)	4 (1, 96)	62 (30, 39)
Wei L. et al. [47]	RCS	Mixed	adult	China	202	116 (57, 43)	23 (11, 39)	13 (6, 44)	4 (1, 98)	n.a.	n.a.
Fu L. et al. [48]	RCS	IP	adult	China	200	99 (49, 5)	109 (54, 5)	120 (60)	n.a.	n.a.	n.a.
Xiaoli X. et al. [49]	RCS	Mixed	pediatric	China	193	120 (62, 18)	120 (62, 18)	23 (11, 92)	8 (4, 15)	15 (7, 77)	32 (16, 58)
Huang M. et al. [50]	RCS	IP	Mixed	China	197	109 (55, 33)	55 (27, 92)	9 (4, 57)	2 (1, 02)	4 (2, 03)	17 (8, 63)
Yan Y. et al. [51]	RCS	IP	adult	China	193	114 (59, 07)	108 (55, 96)	51 (26, 42)	14 (7, 25)	n.a.	68 (35, 23)
Zhou F. et al. [52]	RCS	IP	adult	China	191	119 (62, 3)	53 (27, 75)	9 (4, 71)	7 (3, 66)	n.a.	n.a.
Gong J. et al. [53]	RCS	IP	adult	China	189	88 (46, 56)	28 (14, 81)	8 (4, 23)	n.a.	n.a.	n.a.
CDC MMWR USA [54]	RCS	Mixed	Mixed	USA	180	97 (53, 89)	n.a.	48 (26, 67)	44 (24, 44)	15 (8, 33)	n.a.
Chen D. et al. [55]	RCS	IP	adult	China	175	83 (47, 43)	40 (22, 86)	34 (19, 43)	7 (4)	5 (2, 86)	n.a.
Lu X. et al. [56]	RCS	Mixed	pediatric	China	171	104 (60, 82)	3 (1, 75)	15 (8, 77)	11 (6, 43)	n.a.	n.a.
Garazzino S et al. [57]	RCS	Mixed	pediatric	Italy	168	94 (55, 95)	16 (9, 52)	22 (13, 1)	9 (5, 36)	n.a.	n.a.
Zheng F. et al. [58]	CS	Mixed	adult	China	161	80 (49, 69)	30 (18, 63)	17 (10, 56)	6 (3, 73)	n.a.	n.a.
Mo P. et al. [59]	CS	IP	adult	China	155	86 (55, 48)	37 (23, 87)	7 (4, 52)	3 (1, 94)	3 (1, 94)	26 (31.7)(n.a.)
Yang W. et al. [60]	CS	IP	adult	China	149	81 (54, 36)	14 (9, 4)	11 (7, 38)	2 (1, 34)	n.a.	n.a.
Fan Z. et al. [61]	RCS	IP	adult	China	148	73 (49, 32)	10 (6, 76)	6 (4, 05)	3 (2, 03)	n.a.	n.a.
Chen Q. et al. [62]	RCS	IP	adult	China	145	79 (54, 48)	43 (29, 66)	39 (26, 9)	24 (16, 55)	8 (5, 52)	62 (42, 76)
Feng Z. et al. [63]	RCS	IP	adult	China	141	72 (51, 06)	15 (10, 64)	6 (4, 26)	n.a.	n.a.	5 (3, 55)
Zhang JJ. et al. [64]	CSS	IP	adult	China	140	71 (50, 71)	58 (41, 43)	18 (12, 86)	24 (17, 14)	8 (5, 71)	17 (12, 14)
Wang D. et al. [65]	CS	IP	adult	China	138	75 (54, 35)	36 (26, 09)	14 (10, 14)	14 (10, 14)	3 (2, 17)	55 (39, 86)
Liu K. et al. [66]	RCS	IP	adult	China	137	61 (44, 53)	n.a.	11 (8, 03)	n.a.	n.a.	n.a.
Li X. et al. [67]	RCS	IP	adult	China	131	63 (48, 09)	n.a.	1 (0, 76)	n.a.	n.a.	n.a.
Giacomet V. et al. [68]	RCS	IP	pediatric	Italy	127	83 (65, 35)	20 (15, 75)	28 (22, 05)	12 (9, 45)	8 (6, 3)	n.a.
Bai T. et al. [69]	CSS	IP	adult	China	127	80 (62, 99)	36 (28, 35)	5 (3, 94)	3 (2, 36)	n.a.	n.a.
Wan S. et al. [70]	RCS	IP	adult	China	123	55 (44, 72)	21 (17, 07)	n.a.	n.a.	n.a.	10 (8, 13)
Cholankeril G. et al. [71]	RCS	IP	adult	USA	116	62 (53, 45)	9 (7, 76)	12 (10, 34)	5 (4, 31)	10 (8, 62)	22 (18, 97)
Ma YL. et al. [72]	RCS	Mixed	pediatric	China	115	73 (63, 48)	3 (2, 61)	n.a.	n.a.	n.a.	n.a.
Wang K. et al. [73]	RCS	IP	adult	China	114	58 (50, 88)	n.a.	3 (2, 63)	n.a.	n.a.	n.a.
Klopfenstein T. et al. [74]	RCS	Mixed	adult	France	114	58 (50, 88)	n.a.	55 (48, 25)	25 (21, 93)	19 (16, 67)	n.a.
Peng YD. et al. [75]	RCS	Mixed	adult	China	112	53 (47, 32)	16 (14, 29)	15 (13, 39)	n.a.	n.a.	n.a.
Liu Y. et al. [76]	RCS	IP	adult	China	109	59 (54, 13)	53 (48, 62)	12 (11, 01)	n.a.	n.a.	n.a.
Han R. et al. [77]	CSS	IP	adult	China	108	38 (35, 19)	n.a.	15 (13, 89)	n.a.	n.a.	n.a.
Wang D. et al. [78]	RCS	IP	adult	China	107	57 (53, 27)	20 (18, 69)	7 (6, 54)	6 (5, 61)	2 (1, 87)	n.a.
Zhang H. et al. [79]	RCS	IP	adult	China	107	60 (56, 07)	56 (52, 34)	15 (14, 02)	n.a.	n.a.	n.a.
Tabata S. et al. [80]	RCS	Mixed	adult	Japan	104	54 (51, 92)	28 (26, 92)	9 (8, 65)	n.a.	n.a.	n.a.
Ai J. et al. [81]	RCS	IP	adult	China	102	52 (50, 98)	8 (7, 84)	15 (14, 71)	9 (8, 82)	n.a.	11 (10, 78)
Zhao W. et al. [82]	RCS	IP	adult	China	101	56 (55, 45)	14 (13, 86)	15 (14, 85)	2 (1, 98)	n.a.	n.a.
Chen N. et al. [83]	CS	IP	adult	China	99	67 (67, 68)	17 (17, 17)	n.a.	2 (2, 02)	n.a.	1 (1, 01)
Lin L. et al. [84]	RCS	IP	adult	China	95	45 (47, 37)	20 (21, 05)	23 (24, 21)	21 (22, 11)	2 (2, 11)	17 (17, 89)
Yu X. et al. [85]	RCS	0	adult	China	92	57 (61, 96)	41 (44, 57)	7 (7, 61)	4 (4, 35)	n.a.	n.a.
Qian GQ. et al. [86]	RCS	IP	adult	China	91	37 (40, 66)	9 (9, 89)	21 (23, 08)	11 (12, 09)	n.a.	23 (25, 27)
Zhao XY. et al. [87]	RCS	IP	adult	China	91	49 (53, 85)	30 (33)	14 (15, 38)	19 (20, 88)	n.a.	19 (20, 88)
Xu X. et al. [88]	RCS	IP	adult	China	90	39 (43, 33)	n.a.	5 (5, 56)	5 (5, 56)	n.a.	n.a.
Chen S. et al. [89]	CSS	IP	Mixed	China	89	30 (33, 71)	n.a.	5 (5, 62)	2 (2, 25)	n.a.	n.a.
Xu W. et al. [90]	CSS	IP	Mixed	china	87	46 (52, 87)	40 (45, 98)	n.a.	n.a.	n.a.	n.a.
Kluytmans M. et al. [91]	RCS	Mixed	adult	Germany	86	15 (17, 44)	n.a.	16 (18, 6)	15 (17, 44)	5 (5, 81)	15 (17, 44)
Wei X-S. et al. [92]	CS	IP	adult	China	84	28 (33, 33)	n.a.	n.a.	16 (19, 05)	5 (5, 95)	n.a.
Li K. et al. [93]	RCS	IP	adult	China	83	44 (53, 01)	25 (30, 12)	7 (8, 43)	n.a.	n.a.	n.a.
Shi H. et al. [94]	CS	IP	adult	China	81	42 (51, 85)	n.a.	3 (3, 7)	4 (4, 94)	n.a.	1 (1, 23)
Cai Q. et al. [95]	CSS	IP	adult	China	80	35 (43, 75)	n.a.	1 (1, 25)	n.a.	n.a.	n.a.
Wu J. et al. [96]	RCS	IP	adult	China	80	42 (52, 5)	n.a.	7 (8, 75)	n.a.	n.a.	n.a.
Wu J. et al. [97]	CSS	IP	adult	China	80	39 (48, 75)	3 (3, 75)	1 (1, 25)	1 (1, 25)	n.a.	n.a.
Fang X. et al. [98]	RCS	Mixed	adult	China	79	45 (56, 96)	24 (30, 38)	4 (5, 06)	n.a.	n.a.	5 (6, 33)
Zhao W. et al. [99]	RCS	IP	Mixed	China	77	34 (44, 16)	20 (25, 97)	1 (1, 3)	6 (7, 79)	n.a.	7 (9, 09)
Yu F. et al. [100]	RCS	IP	adult	China	76	38 (50)	17 (22, 37)	3 (3, 95)	4 (5, 26)	n.a.	9 (11, 84)
Zhao Z. et al. [101]	CS	IP	adult	China	75	42 (56)	n.a.	7 (9, 33)	n.a.	n.a.	9 (12)
Wu Y. et al. [102]	RCS	Mixed	adult	China	74	39 (52, 7)	18 (24, 32)	n.a.	n.a.	n.a.	n.a.
Xiao F. et al. [4]	RCS	IP	Mixed	China	73	41 (56, 16)	4 (5, 48)	26 (35, 62)	n.a.	n.a.	n.a.
Tang X. et al. [103]	CCS	IP	adult	China	73	45 (61, 64)	73 (100)	n.a.	n.a.	n.a.	n.a.
Zhou S. et al. [104]	RCS	IP	adult	China	62	39 (62, 9)	n.a.	9 (14, 52)	n.a.	n.a.	n.a.
Xu XW. et al. [105]	RCS	IP	adult	China	62	35 (56, 45)	1 (1, 61)	3 (4, 84)	n.a.	n.a.	n.a.
Miao C. et al. [106]	CSS	IP	adult	China	62	32 (51, 61)	4 (6, 45)	7 (11, 29)	n.a.	n.a.	n.a.
Cheung KS. et al. [107]	RCS	Mixed	adult	China	59	27 (45, 76)	n.a.	13 (22, 03)	1 (1, 69)	7 (11, 86)	n.a.
Grein J. et al. [108]	RCS	IP	adult	USA, Europa and Japan	53	40 (75, 47)	34 (64, 15)	5 (9, 43)	n.a.	n.a.	n.a.
Fu H. et al. [109]	RCS	IP	adult	China	52	28 (53, 85)	10 (19, 23)	7 (13, 46)	1 (1, 92)	n.a.	n.a.
Yang X. et al. [110]	RCS	IP	adult	China	52	35 (67, 31)	52 (100)	n.a.	2 (3, 85)	n.a.	n.a.
Song F. et al. [111]	CSS	IP	adult	China	51	25 (49, 02)	4 (7, 84)	5 (9, 8)	3 (5, 88)	n.a.	9 (17, 65)
Xu T. et al. [112]	RCS	IP	adult	China	51	25 (49, 02)	n.a.	n.a.	n.a.	n.a.	n.a.
Dreher M. et al. [113]	RCS	IP	Mixed	Germany	50	33 (66)	24 (48)	8 (16)	2 (4)	n.a.	n.a.
Xu YH. et al. [114]	CSS	IP	Mixed	China	50	29 (58)	13 (26)	n.a.	n.a.	n.a.	n.a.
Colaneri M. et al. [115]	RCS	Mixed	Mixed	Italy	44	28 (63, 64)	17 (38, 64)	3 (6, 82)	n.a.	n.a.	n.a.
Jin J-M. et al. [116]	CS	IP	adult	China	43	22 (51, 16)	30 (69, 77)	7 (16, 28)	n.a.	n.a.	n.a.
Xiong Y. et al. [117]	RCS	IP	adult	China	42	25 (59, 52)	n.a.	10 (23, 81)	n.a.	n.a.	n.a.
Meng J. et al. [118]	RCS	IP	adult	China	42	24 (57, 14)	16 (38, 1)	3 (7, 14)	2 (4, 76)	n.a.	1 (2, 38)
Huang C. et al. [119]	RCS	IP	adult	China	41	30 (73, 17)	13 (31, 71)	1 (2, 44)	n.a.	n.a.	n.a.
Rodriguez-Lago I. et al. [120]	RCS	IP	adult	Spain	40	24 (60)	2 (5)	9 (22, 5)	n.a.	n.a.	n.a.
Wu WS. et al. [121]	RCS	Mixed	Mixed	China	40	13 (32, 5)	17 (42, 5)	6 (15)	3 (7, 5)	3 (7, 5)	n.a.
Yao N. et al. [122]	CSS	Mixed	adult	China	40	25 (62, 5)	6 (15)	7 (17, 5)	7 (17, 5)	n.a.	n.a.

RCS: retrospective cohort study, CSS: cross-sectional study, ER: epidemiology report, CS: case series, CCS: case control study, MA: meta-analysis, IP: inpatient, OP: outpatient, n.a.: not available

**Table 2 diseases-08-00041-t002:** Summary of the pooled prevalence estimates of gastrointestinal symptoms with respect to whether the study took place in China or out of China.

	Overall		Chinese		Non-Chinese		
	Prevalence (95% CI)	*n*/Patients	Prevalence (95% CI)	*n*/Patients	Prevalence(95% CI)	*n*/Patients	*p* *
Diarrhea	0.15 (0.12, 0.19)	44/62,892	0.12 (0.17, 0.32)	31/12,798	0.24 (0.09, 0.16)	13/50,094	<**0.001**
Nausea/vomiting	0.10 (0.08, 0.12)	44/46,390	0.07 (0.04, 0.10)	29/10,345	0.17 (0.14, 0.19)	15/36,045	<**0.001**
Abdominal pain	0.06 (0.04, 0.07)	38/34,713	0.04 (0.02, 0.05)	22/5272	0.09 (0.07, 0.11)	16/29,441	<**0.001**
Anorexia	0.18 (0.10, 0.27)	37/22,743	0.21 (0.02, 0.51)	31/6099	0.17 (0.11, 0.24)	6/16,644	0.783
Severe	0.18 (0.07, 0.31)	44/75,086	0.17 (0.13, 0.22)	30/11,515	0.17 (0.02, 0.45)	14/63,571	0.978
Any GI symptom	0.21 (0.16, 0.27)	50/71,593	0.19 (0.13, 0.26)	33/13,808	0.24 (0.15, 0.35)	17/57,785	0.370
Fever	0.73 (0.70, 0.76)	45/74,543	0.75 (0.69, 0.81)	30/12,309	0.69 (0.65, 0.73)	15/62,234	0.088
Respiratory	0.60 (0.56, 0.64)	43/74,115	0.55 (0.48, 0.62)	29/12,120	0.70 (0.67, 0.73)	14/61,995	<**0.001**

*n* = number of studies.GI: gastrointestinal; * *p* values were obtained from test for subgroup differences (random effects model). Significant values are shown with bold text.

**Table 3 diseases-08-00041-t003:** Summary of the pooled prevalence estimates of gastrointestinal symptoms with respect to the inpatient/outpatient status.

	Overall		Inpatient		Mix		
	Prevalence (95% CI)	*n*/Patients	Prevalence(95% CI)	*n*/Patients	Prevalence (95% CI)	n/Patients	*p* *
Diarrhea**	0.15 (0.12, 0.19)	44/62,892	0.14 (0.10, 0.19)	24/25,508	0.16 (0.11, 0.21)	18/36,655	0.795
Nausea/vomiting	0.10 (0.08, 0.12)	44/46,390	0.12 (0.09, 0.15)	22/10,228	0.08 (0.05, 0.11)	22/36,162	0.096
Abdominal pain	0.06 (0.04, 0.07)	38/34,713	0.04 (0.03, 0.06)	20/6508	0.07 (0.05, 0.10)	18/28,205	**0.032**
Anorexia	0.18 (0.10, 0.27)	37/22,743	0.15 (0.08, 0.24)	28/19,576	0.27 (0.09, 0.50)	9/3167	0.260
Severe **	0.18 (0.07, 0.31)	44/75,086	0.26 (0.08, 0.49)	24/32,013	0.09 (0.04, 0.16)	20/43,073	0.111

*n* = number of studies.* *p* values were obtained from test for subgroup differences (random effects model).Significant values are shown with bold text. ** The inpatient/outpatient status was not reported in two studies regarding diarrhea and one study regarding severity.

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
