# Peer review of "Newly Reported Studies on the Increase in Gastrointestinal Symptom Prevalence with COVID-19 Infection: A Comprehensive Systematic Review and Meta-Analysis"

_diseases, 2020, doi:10.3390/diseases8040041_

Round 1
Reviewer 1 Report
The authors have reviewed the GI symptoms associated with COVID-19 pandemic. Although this is an interesting piece of work, but the study completely lacks emphasis on GI symptoms on metabolic health. This is important since recent evidences suggest an important role of gut microbiome on human chronic disease. In fact, recent data suggest that alterations in gut microbial population s altered in patients with COVID-19. Moreover, the rate of COVID-19 related morbidity is higher in patients with metabolic conditions such as obesity, diabetes, cardiometabolic problems and liver diseases. In line of chronic metabolic diseases being triggered by GI events such as dysbiosis, mucosal inflammation and barrier dysfunction, it would be interesting to see the link between COVID-19 and GI complications related to chronic diseases.
Author Response
Reviewer #1:
We thank reviewer #1 for his/her comments. The reviewer’s statement about chronic metabolic diseases and triggering of GI events such as dysbiosis, mucosal inflammation and the link of these factors with COVID- 19 GI complications is very important. We thank the reviewer for this suggestion. Although our study design is not suitable for the investigation of this linkage, due to the importance of this topic related to our study we added a whole new paragraph to the discussion section stating the importance of dysbiosis, immune response and the gut-lung axis with six new references.
“Patients with metabolic conditions such as obesity, diabetes, cardio-metabolic problems and liver diseases was repeatedly reported to have higher rates of COVID-19 related morbidity in various studies [134]0.Gut microbiota can influence immune response via affecting the disease progression. Not only over-active and but also hypo-active immune response possibly mediated by the gut microbiota may lead to severe clinical adverse events. The colon includes huge density of bacteria in the families of Bacteroidaceae, Prevotellaceae, Rikenellaceae, Lachnospiraceae and Ruminococcaceae [127], while Bacteroidetes, Firmicutes, and Proteobacteria preponderate in the lung [128]. The gut microbiota may affect pulmonary health through interaction between the gut microbiota and the lungs which is named to as the “gut-lung axis” [129]. The gut-lung axis is reciprocal, so endotoxins, microbial metabolites can affect the lung through the blood and inflammation of the lungs can affect the gut microbiota also [130]. Several studies have shown that respiratory infections are associated with a change in the composition of the gut microbiota [131]. Multiple data suggest that gut microbiota plays a key role in the pathogenesis of sepsis and ARDS. Loss of gut bacterial diversity may lead to dysbiosis that can be associated with many diseases [132]. As many elderly and immune-compromised patients progress to serious adverse clinical consequences in Covid-19, a possible cross-talk may be occurring between lung and intestinal microbiota, which may affect the outcome of clinical manifestation. ”
Reviewer 2 Report
This manuscript represents a systematic literature search on a hot topic affecting today's world. 118 studies were included in the systematic review and 44 were included in meta-analysis specifically looking at GI symptoms in patients with confirmed COVID-19 infection. The methods appear sound as do the conclusions with data that is has been collected in a somewhat erratic way due to the nature of a global pandemic, which the authors discuss. Learning about the other symptoms associated with COVID-19 infection is important for health care providers and patients alike, as not all individuals present with the most common symptoms reported, fever, cough and shortness of breath.
Author Response
Response to reviewers:
Reviewer #2: We thank reviewer #2 for his/her relevant comments.
We thank the reviewer’s first statement indicating that our systematic review and meta- analysis involving 118 and 44 studies respectively. To the best of our knowledge, if this manuscript is accepted that will be highest COVID- 19 patient population included in a study.
In the second statement, the reviewer reminded us that due to the nature of the global pandemic, the related data collected is in a somewhat erratic way but also emphasized that the authors discussed this issue. As the reviewer stated we explained this issue about the data collection and other issues around the scientific publication processes during a disaster period. As the reviewer did not require any further suggestions, we believe that our degree of discussion in the limitations part of the manuscript is enough regarding that statement.
For the third statement of the reviewer; at the reviewer’s suggestion, we emphasized the fact that there is a need to increase the awareness of healthcare providers and patients that not all patients with COVID-19 infection present with classical symptoms such as fever, cough and shortness of breath with the following sentence in the conclusion section:
“Although healthcare providers and patients are well aware of the common symptoms of COVID-19 such as fever, cough, and shortness of breath, there is need to raise the awareness about the fact that not all individuals present with them and gastrointestinal symptoms are among relatively common symptoms associated with this disease.”
Reviewer 3 Report
Dear Authors and the team,
I had the pleasure in reading the manuscript. Please pay attention to the English grammar in certain places and spelling. It is an interesting study. I think it will add to the knowledge of this field.
Best wishes,
S.S.
Author Response
Reviewer #3:
We are very thankful to reviewer #3 for his/her comments about having the pleasure to read the manuscript and stating that the study is interesting and will add to knowledge in this field. For the suggestion of paying attention to English grammar in certain places, although we had two different native English speaker controls before the manuscript sent to the journal, an additional thorough native speaker control has been completed. We responded to these queries in a point-by-point manner and made the changes in the text using the “track changes” option. We hope these will fulfill the requirements.
Round 2
Reviewer 1 Report
In the revised version of the manuscript, the authors have satisfactorily addressed the concerns related to COVID-19 associated co-morbidities.